# OpenReview forum: "MRAG-Corrupter: Knowledge Poisoning Attacks to Multimodal Retrieval Augmented Generation"
_ICLR.cc/2026/Conference — Submitted to ICLR 2026_

### Official Review · Reviewer_ynM3 · 2025-10-21

**Soundness:** 3
**Presentation:** 3
**Contribution:** 3
**Rating:** 6
**Confidence:** 3

**Summary:**

This paper studies the security vulnerabilities of multimodal retrieval augmented generation. To bridge the gap on attack on multi-modal systems, the paper proposes MRAG-Corrupter, the first knowledge poisoning attack tailored for multimodal RAG. It derives two cross-modal solutions and evaluates it across multiple knowledge databases and victim VLMs. The attack method outperforms baseline methods. Various deffense strategies are also investigated.

**Strengths:**

- The topic studied in this paper is important. In particular, it focuses on multimodal RAG systems, which is under-explored now.
- The proposed MRAG-Corrupter demonstrated consistent performance improvements over exisiting methods and achieved high ASRs. A set of powerful models are taken as the victim VLMs, which makes the conclusion more generalizable.
- Two retriever access scenarios, restricted-access and full-accsess, are considered. Attack strategies are designed for the corresponding ones.
- Overall, the paper structure is clear and figures are nicely presented.

**Weaknesses:**

- In Sec. 2 Problem Formulation and Sec. 3 MRAG-Corrupter, the authors introduced a lot of notations, it is a little bit easy to get lost to follow them. E.g., the use of \dot over letters in Ln 96.

**Questions:**

- In Ln 180, the heuristic approach directly injects the query image $\dot{I}_i$ as $\tilde{I}_i^j$ while keeping $\tilde{T}_i^j$ unchanged. Why not do it reversely, i.e., inject the text while keeping image unchanged?

---

> ### Author Response · Authors · 2025-11-21
>
> Dear Reviewer ynM3,
>
> Thank you very much for your careful reading and constructive remarks. Our detailed replies to all points are given below.
>
> ---
>
> ## Q1: Some notations are difficult to follow.
>
> **R1:**
> We thank the reviewer for the comment regarding notations. We acknowledge that some of the symbols in Sec. 2 and Sec. 3, such as the use of the dot notation, may be difficult to follow. We will clarify the notations and improve readability in the revised version.
>
> ---
>
> ## Q2: Why not keep the image unchanged and inject only the text?
>
> **R2:**
> Regarding the heuristic approach mentioned in Line 180: we have explored the alternative strategy of injecting only the text while keeping the image unchanged (the base image). Please refer to **Table 3** under the setting **“Base w. Q”**. The results show that this variant is *less effective* than our main attack setting.
>
> ---

---

### Official Review · Reviewer_2bGb · 2025-10-27

**Soundness:** 3
**Presentation:** 3
**Contribution:** 3
**Rating:** 6
**Confidence:** 3

**Summary:**

The authors propose MRAG-Corrupter, a targeted knowledge poisoning attack on multimodal RAG systems, combining a retriever with a VLM. The attack enables an adversary to control VLM outputs for chosen queries by injecting a small number of carefully crafted image-text pairs into the knowledge base. MRAG-Corrupter combines embedding similarity optimization, surrogate-based generative text refinement, and either direct copying (dirty-label) or stealthy perturbation (clean-label) of images to create “retrieval-dominant” and “generation-dominant” poisoned pairs. These pairs are highly likely to be retrieved and used by the VLM for target queries, thus allowing an attacker to control outputs with minimal, well-crafted database modifications.

**Strengths:**

The paper is clear and easy to follow. It proposes a novel method to attack RAG systems. The authors rigorously formalize the attack as an optimization problem and introduce two cross-modal strategies tailored for varying attacker knowledge and access levels. The authors also provide rigorous experiments, evaluating over leading VLMs and adapted baselines.

**Weaknesses:**

My primary concern pertains to the practical deployment of the proposed MRAG-Corrupter method. While the attack framework assumes access to a well-structured knowledge base comprising image-text pairs, real-world settings commonly employ access controls, provenance tracking, and other integrity mechanisms that may substantially reduce the risk of knowledge poisoning. Additionally, the community may be more interested in attacks targeting online, less curated, or dynamic multimodal knowledge sources rather than static databases. The experimental evaluation is limited to image-text pairs, without consideration of other relevant modalities such as tables, videos, or audio. It remains unclear how transferable the proposed attack strategies and the underlying optimization logic would be to these alternative modalities.

**Questions:**

I encourage the authors to discuss the generalizability of their method and present evidence or analysis regarding its applicability to different types of multimodal knowledge bases.

---

> ### Author Response · Authors · 2025-11-21
>
> # Dear Reviewer 2bGb
>
> We sincerely appreciate your thoughtful review and suggestions. Please find our detailed responses to all comments below.
>
> ---
>
> ## Q1: Real-world settings commonly employ access controls, provenance tracking, and other integrity mechanisms that may substantially reduce the risk of knowledge poisoning.
>
> **R1:**
> We agree that strong access control and provenance tracking can reduce the risk of poisoning in some deployments. However, many practical multimodal RAG systems rely on large, partially curated, or user-contributed corpora—such as Infoseek—which inherently lack such strict governance. Our experiments, including four defenses and additional CLIP-based and MLLM-based reranking, consistently show that the attack remains effective under these more realistic and imperfect conditions. We will clarify this limitation and note provenance-aware defenses as an important direction for future work.
>
> ### **CLIP similarity detection**
>
> | Dataset / Method   | Avg. Image–Text Similarity ↑ |
> | ------------------ | ---------------------------- |
> | Clean database     | 0.2902                       |
> | Ours (dirty-label) | 0.2502                       |
> | Ours (clean-label) | 0.3169                       |
>
> ### **Multimodal reranking**
>
> #### *m = 5*
>
> | Method      | ASR-R  | ASR-G | ACC  |
> | ----------- | ------ | ----- | ---- |
> | Dirty-label | 1.0000 | 0.98  | 0.02 |
> | Clean-label | 0.9467 | 0.86  | 0.06 |
>
> #### *m = 10*
>
> | Method      | ASR-R  | ASR-G | ACC  |
> | ----------- | ------ | ----- | ---- |
> | Dirty-label | 0.3667 | 0.34  | 0.42 |
> | Clean-label | 0.4133 | 0.38  | 0.54 |
>
> ---
>
> ## Q2: The community may be more interested in attacks targeting online, less curated, or dynamic multimodal knowledge sources rather than static databases.
>
> **R2:**
> We agree that attacks on online, less-curated, or dynamic sources are highly relevant. While Infoseek is a static, Wikipedia-derived corpus, it intentionally reflects real-world noise and imperfect alignment typical of web-scraped or user-contributed data. More importantly, the core attack mechanism relies on similarity-based retrieval and manipulable embedding spaces, which are also fundamental to many dynamic/online pipelines; therefore we expect the attack to transfer to those settings (e.g., via injected user uploads). We will clarify this point in the revision and plan to evaluate the method on dynamic, less-curated corpora in future work.
>
> ---
>
> ## Q3: The experimental evaluation is limited to image-text pairs, without consideration of other relevant modalities. It remains unclear how transferable the proposed attack strategies and the underlying optimization logic would be to these alternative modalities.
>
> **R3:**
> We thank the reviewer for this insightful comment. While our experiments focus on image–text RAG, this modality is currently the most widely deployed in both daily applications (e.g., shopping assistants, web search, content recommendation) and specialized domains (e.g., medical image retrieval). To avoid confusion with broader multimodal RAG settings that include audio or video, we can revise the terminology in the paper to use a more precise descriptor (e.g., *image–text retrieval*). We also acknowledge the importance of exploring additional modalities and plan to extend our study to multimodal RAG systems involving audio, video, or tables in future work. Importantly, the underlying attack principle—optimizing inputs to manipulate embedding similarity and satisfy retrieval and generation conditions—is largely modality-agnostic; thus, similar attacks may extend to future systems that rely on embedding-based retrieval with cross-modal encoders.
>
> ---

---

> ### Comment · Reviewer_2bGb · 2025-11-25
>
> Thank the authors for clarification. I have reviewed the authors' response and the updated manuscript. I would like to keep my rating to show my support of this paper.

---

### Official Review · Reviewer_jEaj · 2025-10-31

**Soundness:** 3
**Presentation:** 3
**Contribution:** 1
**Rating:** 2
**Confidence:** 5

**Summary:**

This paper proposes MRAG-corrupter, a knowledge poisoning attack on vision-language RAG systems. It aims to manipulate the MRAG system's response to specific target queries by injecting poisoning samples into the knowledge database. The paper formulates the attack as an optimization problem and discusses to solve it in two different settings (i.e., restricted-access and full-access setting) depending on the attacker's knowledge. Extensive experiment results demonstrate that the proposed attack is both effective and robust.

**Strengths:**

•	This paper have a good writing to express the key idea clearly. The figures and tables are also clear and easy to understand.
•	The paper includes comparison with many baseline methods, demonstrating the effectiveness of the proposed attack.
•	The paper discusses many possible defenses to demonstrate the robustness of the attack.

**Weaknesses:**

•	The core idea of this paper is similar to an existing work [1] on corrupting vision-language RAG systems, but [1] is neither cited nor compared in this paper.
•	The paper only discusses MRAG systems that retrieve images and texts. However, MRAG may not limit to images and texts. It can also support other modalities (e.g., audio [2]) and retrieval scenarios [3]. Therefore, the use of MRAG may not be appropriate here.
•	The paper lacks exploration of practical real-world scenarios for the attack.
[1] PoisonedEye: Knowledge Poisoning Attack on Retrieval-Augmented Generation based Large Vision-Language Models. ICML 2025.
[2] WavRAG: Audio-Integrated Retrieval Augmented Generation for Spoken Dialogue Models. ACL 2025.
[3] UniIR: Training and Benchmarking Universal Multimodal Information Retrievers. ECCV 2024.

**Questions:**

•	What are the main differences between your work and existing work (e.g., [1])? Could you further compare experiment results with [1] to demonstrate the difference?
•	In the clean label attack, why the optimized image still preserves semantic meaning to texts? Instead of human reviewers, is it detectable by an embedding model like CLIP?
•	Since the experiments only use CLIP-based models, could the authors conduct more experiments on MLLM-based retrievers like GME [4]?
•	For robustness, is the attack robust to multimodal reranking [5] and defenses such as RoCLIP [6]?
•	Could the authors provide any experiments or examples of real-world applications of the attack, such as compromising a RAG-based LLM agent?
[4] GME: Improving Universal Multimodal Retrieval by Multimodal LLMs. arXiv 2024.
[5] MLLM Is a Strong Reranker: Advancing Multimodal Retrieval-augmented Generation via Knowledge-enhanced Reranking and Noise-injected Training. arXiv 2024.
[6] Robust contrastive language-image pretraining against data poisoning and backdoor attacks. NeurIPS 2023.

---

> ### Author Response · Authors · 2025-11-21
>
> Dear Reviewer jEaj,
>
> We greatly appreciate your insightful comments and constructive feedback. Our point-by-point responses are provided below.
>
> ---
>
> ## Q1: Main difference and results comparison to PoisonedEye.
>
> **R1:**
> We appreciate the reviewer for pointing us to this related work. We will include an appropriate citation and comparisons in the revised version.
>
> The main differences between our work and PoisonedEye are as follows:
> (1) PoisonedEye focuses solely on dirty-label attacks, since the injected image–text pairs are intentionally mismatched. In contrast, our work introduces a clean-label attack setting, where the injected image–text pairs remain semantically aligned.
> (2) PoisonedEye generates adversarial texts using a prompt-injection technique. In our approach, adversarial texts are produced through iterative refinement with an LLM and concatenation with the target query text, leading to more controlled and semantically consistent crafted texts.
>
> We conducted additional experiments to compare the attack effectiveness of PoisonedEye with that of our method under our main experimental settings. The results are reported in the tables below. The results show that our dirty-label attack achieves slightly higher ASRs than PoisonedEye.
>
> ### **Infoseek**
>
> | Method             | ASR-R  | ASR-G | ACC  |
> | ------------------ | ------ | ----- | ---- |
> | PoisonedEye-B      | 0.9000 | 0.90  | 0.10 |
> | PoisonedEye-S      | 0.9000 | 0.90  | 0.10 |
> | Ours (Dirty-label) | **1.00**   | **0.98**  | **0.02** |
>
> ### **OVEN**
>
> | Method             | ASR-R  | ASR-G | ACC  |
> | ------------------ | ------ | ----- | ---- |
> | PoisonedEye-B      | 0.9867 | 0.94  | 0.02 |
> | PoisonedEye-S      | 0.9933 | 0.94  | 0.02 |
> | Ours (Dirty-label) | **1.00**   | **0.96**  | **0.02** |
>
> However, although the two attacks achieve comparable results, the distinctions between our method and PoisonedEye make our attack substantially stealthier against detection mechanisms. Specifically: (1) CLIP similarity detection, where a CLIP model (ViT-H/14) is used to evaluate image–text embedding similarity; and (2) prompt-injection detection, where tools such as Prompt Armor or the DataSentinel model are applied to identify potential prompt-injection patterns in the injected texts. The results indicate that our attack is significantly more covert than PoisonedEye.
>
> * **CLIP similarity detection** using CLIP (ViT-H/14)
> * **Prompt-injection detection** using tools such as Prompt Armor or DataSentinel
>
> ### **CLIP similarity detection**
>
> | Dataset / Method   | Avg. Image–Text Similarity ↑ |
> | ------------------ | ---------------------------- |
> | Clean database     | 0.2902                       |
> | PoisonedEye-B      | 0.1842                       |
> | PoisonedEye-S      | 0.2081                       |
> | Ours (Dirty-label) | 0.2502                       |
> | Ours (Clean-label) | **0.3169**                   |
>
> ### **Prompt-injection detection (TPR)**
>
> | Method      | Prompt Armor ↓ | DataSentinel ↓ |
> | ----------- | -------------- | -------------- |
> | PoisonedEye | 0.870          | 0.860          |
> | Ours        | **0.000**          | **0.044**          |
>
> ---
>
> ## Q2: In the clean label attack, why the optimized image still preserves semantic meaning to texts? Instead of human reviewers, is it detectable by an embedding model like CLIP?
>
> **R2:**
> (1) We have provided examples of perturbed images in Figure 13 and reported the LPIPS scores in our response to Reviewer sj9S, which demonstrate that the perturbed images remain highly similar to the originals and therefore preserve the semantic consistency with their corresponding texts.
> (2) We also showed that the clean-label attack yields an average CLIP similarity of 0.3169, compared to 0.2902 for the clean knowledge database, indicating that the attack is difficult to detect using embedding-based models such as CLIP.
>
> ---
>
> ## Q3: Experiments on MLLM-based retrievers like GME.
>
> **R3:**
> We conducted experiments using GME on the Infoseek knowledge database, and the results are presented in the table below. The findings show that our attack remains effective under this setting. It is also worth noting that we used our perturbed images from the main clean-label attack experiments and observed that the transferred performance on the GME model is similarly strong.
>
> | Method                      | ASR-R  | ASR-G | ACC  |
> | --------------------------- | ------ | ----- | ---- |
> | Ours (Dirty-label)          | 0.8667 | 0.88  | 0.04 |
> | Ours (Clean-label transfer) | 0.8467 | 0.82  | 0.08 |
>
> ---

---

> ### Author Response · Authors · 2025-11-21
>
> ## Q4: For robustness, is the attack robust to multimodal reranking and defenses such as RoCLIP?
>
> **R4:**
> We conducted experiments using multimodal reranking based on GPT-4o, where we reranked the top-k retrieved candidates (m = 5, 10), and the results are shown below. When m = 5, the attack is barely mitigated. For m = 10, both ASR-R and ASR-G decrease, yet remain above 30%, while ACC drops by more than half, indicating that multimodal reranking alone is insufficient to defend against our attack.
>
> ### **Multimodal reranking (m = 5)**
>
> | Method      | ASR-R  | ASR-G | ACC  |
> | ----------- | ------ | ----- | ---- |
> | Dirty-label | 1.0000 | 0.98  | 0.02 |
> | Clean-label | 0.9467 | 0.86  | 0.06 |
>
> ### **Multimodal reranking (m = 10)**
>
> | Method      | ASR-R  | ASR-G | ACC  |
> | ----------- | ------ | ----- | ---- |
> | Dirty-label | 0.3667 | 0.34  | 0.42 |
> | Clean-label | 0.4133 | 0.38  | 0.54 |
>
> Although performance decreases at k=10, both ASR-R and ASR-G remain above 30%, while ACC drops by more than half — showing that reranking is insufficient.
>
> We also applied the RoCLIP strategy, following the implementation in PoisonedEye, which uses CLIP to rematch each image to the text with the highest similarity score. The results show that while RoCLIP can partially defend against dirty-label attacks—due to the potential mismatch between images and texts—it is much less effective against clean-label attacks. Additionally, the no-attack ACC decreased by 0.12, indicating a notable utility trade-off.
>
> ### **RoCLIP**
>
> | Setting              | ASR-R  | ASR-G | ACC  |
> | -------------------- | ------ | ----- | ---- |
> | w/o defense          | -      | -     | 0.96 |
> | w/o attack + RoCLIP  | -      | -     | 0.84 |
> | Dirty-label + RoCLIP | 0.9333 | 0.18  | 0.70 |
> | Clean-label + RoCLIP | 0.3067 | 0.38  | 0.42 |
>
> ---
>
> ## Q5: MRAG may not limit to images and texts. It can also support other modalities (e.g., audio) and retrieval scenarios. Therefore, the use of MRAG may not be appropriate here.
>
> **R5:**
> We thank the reviewer for pointing out the terminology issue. We agree that *MRAG* is a broader term that encompasses audio, video, and other multimodal retrieval settings, while our work focuses specifically on the image–text scenario. To avoid confusion, we can revise the terminology in the paper to use a more precise term (e.g., “image–text retrieval”). As for different retrieval scenarios mentioned in UniIR, we have discussed the image-pair, text-pair and merge retrieve strategies in the structure-driven mitigation section. In addition, we will add a brief discussion on the applicability of our attack framework to other modalities, which we view as an interesting direction for future work.
>
> ---
>
> ## Q6: The paper lacks exploration of practical real-world scenarios for the attack.  Could the authors provide any experiments or examples of real-world applications of the attack, such as compromising a RAG-based LLM agent?
>
> **R6:**
> We appreciate the reviewer’s suggestion regarding real-world scenarios. Our experiments are conducted on a Wikipedia-based multimodal dataset, which naturally contains noise, heterogeneous image quality, and diverse text descriptions, making it representative of real-world MRAG retrieval conditions. Moreover, our attack can directly transfer to practical applications; for instance, a RAG-based shopping assistant that retrieves similar products from a multimodal catalog could be manipulated to recommend an attacker-preferred brand by retrieving an injected adversarial image–text pair instead of the legitimate match.

---

> > ### Comment · Reviewer_jEaj · 2025-11-26
> >
> > Thanks for thoroughly addressing my comments. I have accordingly raised my score.
> >
> >
> >
> > In my opinion, this paper contributes in: 1) the introduction of iterative refinement process for poison text generation; 2) the use of DALL·E to generate the base poison image to align the image with poison text. These two techniques improved the stealthiness of the injected image-text pair to evade the detection methods like similarity-based detection and prompt-injection detection. I encourage the authors emphasize this improvement in stealthiness as a key contribution in the paper.
> >
> > Besides, it should be noted that the image–text RAG poisoning setting and the optimization of the perturbation δ have been explored in prior work.
> >
> >
> >
> > For robustness, it is quite concerning that the ASR drops to 30%-40% in multimodal reranking with m=10. Since reranking (usually with a larger m, e.g., m=20 in [5]) is a common component in many RAG pipelines to improve accuracy, this performance degradation raises questions about the practical effectiveness of the proposed attack in real-world deployments.
> >
> >
> >
> > Minor: In the revised paper, the table number in line 333 and 482 is missing

---

> > > ### Author Response · Authors · 2025-11-28
> > >
> > > We thank the reviewer for carefully reading our rebuttal and revised manuscript, and for the constructive comments and suggestions — we truly appreciate the time and effort, as well as the reviewer’s willingness to reconsider the score. We have revised the paper to better clarify the connection to prior work on attacking image–text RAG systems (e.g., *PoisonedEye*) and to highlight the stealthiness of our attack (line 44–48 and line 80).
> > >
> > > To further evaluate robustness, we rerank the top-20 retrieved candidates using a multimodal LLM. The results are shown in the table below:
> > >
> > > | Setting | ASR-R | ASR-G | ACC |
> > > |--------|-------|--------|------|
> > > | **m = 20 (dirty-label)** | 0.3333 | 0.20 | 0.64 |
> > > | **m = 20 (clean-label)** | 0.4067 | 0.36 | 0.58 |
> > >
> > > We observe that the ASR of dirty-label attacks is lower than that of clean-label attacks. A potential reason is that the MLLM-based reranker can detect semantic inconsistencies between images and texts in the dirty-label setting, thereby partially filtering out mismatched pairs. In contrast, clean-label poisoning preserves image–text alignment, making anomalies more difficult for the reranker to identify. Importantly, even under the challenging *m = 20* condition, the ASR remains around 0.3, showing that injecting only a small number of malicious image–text pairs into a large knowledge base (481k pairs in our experiments) can still meaningfully influence retrieval and generation, posing a non-trivial security risk.
> > >
> > > We hope the reviewer finds our clarifications and newly added experiments helpful in better understanding the contributions of our work. In particular, we introduce:
> > >
> > > 1. **A novel and stealthy attack formulation** that remains effective while preserving perceptual robustness.
> > > 2. **A comprehensive evaluation across diverse defense strategies**, which we believe may offer meaningful insights for strengthening future image–text RAG systems.
> > >
> > > Our intention is to raise awareness of this underexplored class of stealthy attacks and to encourage the development of more practical and robust defenses. We sincerely appreciate the reviewer’s time and thoughtful feedback, and we would be grateful if the additional results help address the concerns and allow for a reconsideration of the score.

---

### Official Review · Reviewer_sj9S · 2025-11-01

**Soundness:** 3
**Presentation:** 4
**Contribution:** 3
**Rating:** 6
**Confidence:** 3

**Summary:**

This paper introduces MRAG-Corrupter, a knowledge poisoning attack framework specifically designed for Multimodal RAG systems. Multimodal RAG enhances VLMs by allowing them to retrieve information from external knowledge bases containing images and text. The authors identify a security vulnerability: by injecting a small number of maliciously crafted image-text pairs into the knowledge database, an attacker can manipulate the VLM to generate attacker-desired, incorrect answers to specific user queries.
The attack is formalized as an optimization problem with two necessary conditions: a Retrieval Condition (ensuring the malicious pairs are retrieved) and a Generation Condition (ensuring the VLM produces the target answer from the retrieved pairs). To address different threat scenarios, the authors propose two attack strategies: (1) Dirty-Label Attack: For a restricted-access scenario where the attacker has no access to the retriever. It directly reuses the query image and prepends the query text to a maliciously generated description. (2) Clean-Label Attack: For a full-access scenario where the attacker can query the retriever. It generates semantically aligned image-text pairs and adds subtle perturbations to the image to maximize retrieval similarity while appearing legitimate to human moderators.

**Strengths:**

+ Clear Illustration
+ Technically sound design
+ Comprehensive evaluation

**Weaknesses:**

1. I am curious that if a knowledge base already contains an image–text pair that exactly matches a query, how can an attacker craft a new pair that the VLM retrieves and uses to produce the attacker’s desired output. Is this vulnerability primarily due to encoder weaknesses or to retrieval and ranking dynamics?

2. While the paper evaluates four defense strategies, the defenses are tested in isolation, but real-world systems may employ multiple complementary defense. It would be better to discuss this aspect.

3. The clean-label attack requires DALL·E-3 to generate aligned image-text pairs and assumes access to the retriever in full-access scenarios. This creates a significant barrier for real-world deployment where such access might be restricted. Moreover, the clean-label attack's effectiveness is tied to perturbation intensity ε, with higher values (ε = 32/255). Though the author claims that it remains stealthy, but it is better to showcase more concrete examples. It seems there is a practical trade-off in stealth versus effectiveness.

**Questions:**

1. It is suggested to clarify the foundamental reason for such vulnerability.

2. It is better to consider and discuss compound defenses.

3. It is better to analyze the practical trade-off between stealth and effectiveness for clean-label attack.

---

> ### Author Response · Authors · 2025-11-21
>
> Dear Reviewer sj9S，
>
> Thank you for your valuable feedback and thoughtful suggestions. We have carefully addressed each comment, as detailed below.
>
> ---
>
> ## Q1: Can an attacker craft an image-text pair if the knowledge database contains a pair that exactly matches a query? What is the fundamental reason for such vulnerability?
>
> **R1:**
> From our understanding, if a knowledge base already contains exactly the user query, it’s certainly harder (or even impossible) to achieve the retrieval condition for attacker. However, if the text in the database is a query, then the data simply doesn’t provide any external knowledge, defeating the purpose of RAG systems. In realistic settings, the database may contain related but not identical image–text pairs to the queries.
>
> Our results show that attackers can craft adversarial pairs whose embeddings are closer to the query than the benign candidates. This happens due to:
>
> 1. **Encoder weaknesses** — multimodal encoders can be manipulated to shift embeddings through adversarial images and crafted captions.
> 2. **Retrieval/ranking dynamics** — retrieval relies purely on embedding similarity and therefore ranks the adversarial pair higher.
>
> It is worth noticing that, while our paper we mainly focus on CLIP similarity score-based ranking, we have added an experiment to show that MLLM-based reranking is also insufficient for defending our attack. We have used GPT-4o for reranking the top-m (m = 5, 10) retrieved candidates; the results are shown below.
> ### **Multimodal reranking (m = 5)**
>
> | Method      | ASR-R  | ASR-G | ACC  |
> | ----------- | ------ | ----- | ---- |
> | Dirty-label | 1.0000 | 0.98  | 0.02 |
> | Clean-label | 0.9467 | 0.86  | 0.06 |
>
> ### **Multimodal reranking (m = 10)**
>
> | Method      | ASR-R  | ASR-G | ACC  |
> | ----------- | ------ | ----- | ---- |
> | Dirty-label | 0.3667 | 0.34  | 0.42 |
> | Clean-label | 0.4133 | 0.38  | 0.54 |
>
> ---
>
> ## Q2: It is better to consider and discuss combined defenses.
>
> **R2:**
> Thank you for the suggestion! We note that the currently discussed defenses often incur substantial utility trade-offs. For example, the structure-driven mitigation (Figure 6) shows that the no-attack ACC and ASR-G are positively correlated, and the purification defense reduces the no-attack ACC by 0.12. Therefore, we believe it is more appropriate to present the defense results separately.
>
> For paraphrasing and duplicate removal, the combined defense results for Infoseek are provided in the table below. The results still indicate that these defenses remain insufficient for mitigating our attack.
>
> | Method      | ASR-R | ASR-G | ACC  |
> | ----------- | ----- | ----- | ---- |
> | Clean-label | 0.79  | 0.74  | 0.20 |
> | Dirty-label | 0.33  | 0.54  | 0.42 |
>
> ---
>
> ## Q3: The trade-off between stealth and the effectiveness of the clean-label attack
>
> **R3:**
> We provide ablation studies reporting ASRs under epsilons of 8/255 and 16/255 in Figure 4, and we additionally include visual examples of perturbed images across different ε constraints in Figure 13 in Appendix. To further quantify perceptual similarity, we report LPIPS scores distribution in Figure 7 (the revised version of  our paper) showing that the perturbations introduce only minimal changes: even at ε = 32/255, over 75% of images have LPIPS differences below 0.04.
>
> We acknowledge that direct access to the retriever may be restricted in some application scenarios; in such cases, we propose the dirty-label attack as a practical alternative strategy.
>
> ---

---

> > ### Comment · Reviewer_sj9S · 2025-11-26
> > **Reply**
> >
> > Thank the authors for clarification. I have reviewed the authors' response and the updated manuscript. I will keep my current rating.

---

### Author Response · Authors · 2025-11-21
**Summary of Paper Revision**

# Summary of Paper Revisions

We thank all reviewers for their constructive feedback and have made the following updates:
1. **Introduction** (Pages 1-2): Cited *PoisonedEye*, explained our work's connection with prior work and emphasized attack stealthiness as a contribution.
1. **Main Results** (Pages 6–7): Added *PoisonedEye* as a baseline, and included CLIP-based and prompt injection detection.
2. **Ablation Studies** (Page 8): Measured LPIPS differences for clean-label attack images and added experiments using the MLLM-based retriever GME.
3. **Defense Analysis** (Page 9): Expanded evaluation to include multimodal reranking and RoCLIP as defense strategies.
4. **Related Work** (Page 10): Added discussion of *PoisonedEye* as related work.
5. **Future Work** (Page 10): Updated future directions to emphasize exploration of other modalities and less curated datasets.
6. **Appendix E.3** (Page 19): Added details of the *PoisonedEye* baseline.
7. **Appendix E.4** (Page 19): Added detailed CLIP-based and prompt injection detection results for *PoisonedEye* and our attack.
8. **Appendix E.5** (Page 20): Added detailed LPIPS results for the clean-label attack.
9. **Appendix F.3** (Page 22): Added detailed RoCLIP defense results.

---

### Comment · Area_Chair_Hymw · 2025-11-25
**Post-Rebuttal Discussion**

Dear all,

Could you first review the original comments from other reviewers and the rebuttal materials, and then post your comments? Discussion is necessary for this paper.

Best,
AC

---

### Author Response · Authors · 2025-12-02
**Summary of Reviews and Rebuttal**

Dear AC,

Thank you very much for handling our paper. Below, we summarize the reviews and describe how the revised paper addresses each of the comments.

### **Strengths**
1. Importance and timeliness of the topic (Reviewer ynM3)
2. Sound and novel technical design (Reviewer sj9S, Reviewer 2bGb, Reviewer ynM3)
3. Comprehensive experimental evaluation (Reviewer sj9S, Reviewer jEaj, Reviewer 2bGb, Reviewer ynM3)
4. Clarity of writing and presentation (Reviewer sj9S, Reviewer jEaj, Reviewer 2bGb, Reviewer ynM3)

### **Weaknesses**
1. Writing clarifications

    a. Clarity of assumptions and fundamental reason for the vulnerability (Reviewer sj9S)
    b. Relation to prior work and completeness of citation (Reviewer jEaj)
    c. Scope of MRAG definition and modality coverage (Reviewer jEaj, Reviewer 2bGb)
    d. Real-world applicability and deployment scenarios (Reviewer jEaj, Reviewer 2bGb)
    e. Complexity of notation and clarification of design choices (Reviewer ynM3)

2. Additional experiments

   a. Differences from and comparisons with PoisonedEye (Reviewer jEaj)

   b. Detectability of clean-label perturbations by embedding-based detectors (e.g., CLIP) (Reviewer jEaj)

   c. Experiments using MLLM-based retrievers such as GME (Reviewer jEaj)

   d. Robustness to multimodal reranking and defenses such as RoCLIP (Reviewer jEaj)

   e. Evaluation of combined defenses (Reviewer sj9S)

   f. Clean-label attack practicality and the stealth–effectiveness trade-off (Reviewer sj9S)

### **Our revisions to address the weaknesses**
As the review summary above shows, the weaknesses largely concern clarifications in writing, discussions of non-essential components, and a few small additional experiments. In contrast, the problem studied and the proposed method are novel. Below, we provide detailed explanations of how we addressed each of the stated weaknesses.

1. We analyzed the fundamental reason for the vulnerability; cited PoisonedEye and clarified our attack’s difference from it;   suggested changing the multimodal RAG terminology to vision-language RAG or image-text RAG; added examples illustrating real-world attack impact and deployment relevance; and clarified notation and design choices.

2. We added comparisons against PoisonedEye and included additional experiments involving CLIP-based and prompt-injection detectors, the GME retriever, multimodal reranking, and RoCLIP.  In addition, we evaluated combined defense effectiveness of paraphrasing and duplicates removal;  and evaluated LPIPS differences of the perturbed images and clarified the clean-label attack’s effectiveness under low-perturbation settings.


Reviewer **sj9S** and **2bGb** have reviewed our rebuttal, agreed that we adequately addressed their concerns, and have therefore decided to maintain their positive ratings of 6 in support of our paper.

Reviewer **jEaj** agrees that we have addressed the comments and raised the score from 2 → 4. The reviewer further suggested evaluating multimodal reranking with top-20 candidates; we conducted this experiment and found that our attack continues to meaningfully influence retrieval and generation under this setting. However, the reviewer was unable to continue the discussion because the rebuttal period had already frozen.

Reviewer **ynM3** was unable to participate during the rebuttal phase, but his/her rating of 6 was already positive.

Thank you very much for your time!

Thanks,

Authors

---

### Meta-Review · Area_Chair_buZD · 2025-12-23

**Summary:**

This paper introduces MRAG-Corrupter, a knowledge poisoning attack against Multimodal RAG systems. MRAG-Corrupter contains two strategies: dirty-label and clean-label. Experimental results demonstrate the effectiveness of MRAG-Corrupter. However, the feasibility of poisoning attacks on MRAG has been discussed by PoisonedEye, and MRAG-Corrupter only represents an incremental improvement over PoisonedEye. Meanwhile, the methodology of MRAG-Corrupter is straightforward and lacks theoretical analysis. Furthermore, the attack settings do not align with real-world scenarios (Reviewer sj9S, Reviewer jEaj, and Reviewer 2bGb), and this paper did not conduct experiments on real-world Multimodal RAG systems (as introduced in Section 6). It is recommended that the authors analyze the fundamental reasons for such vulnerability and discuss them within a broader range of real-world scenarios in future research.

**Reviewer Concerns:**

Addressed Concerns:
- Discuss the main difference and results comparison to PoisonedEye.
- Discuss the robustness of MRAG-Corrupter on more advanced defenses.

Unaddressed Concerns:
- Delve into the fundamental reason for such vulnerability.
- Experiments on real-world scenarios.
- The generalization of MRAG-Corrupter on alternative modalities.

**Reviewer Scores:**

In my view, certain review concerns remain unaddressed, and the score should be lowered after thorough discussion.

---

### Decision · Program_Chairs · 2026-01-26

Reject